# Gendered Associations between Single Parenthood and Child Behavior Problems in the United Kingdom

**DOI:** 10.3390/ijerph192416726

**Published:** 2022-12-13

**Authors:** Samuel C. M. Faulconer, M. Rachél Hveem, Mikaela J. Dufur

**Affiliations:** Department of Sociology, Brigham Young University, Provo, UT 84602, USA

**Keywords:** family structure, single parent, single fathers, internalizing behavior problems, externalizing behavior problems

## Abstract

Internalizing and externalizing behavior problems are associated with a variety of negative child outcomes, but these conclusions have been drawn from research that usually compares children in families with two biological, married parents to all other family types. We compare behavior problems across two-parent, single-mother, and single-father families, which allows us to explore competing gender theories as possible explanations for why child behavior outcomes may be different across these three categories. Results from analyses of the UK Millennium Cohort Study suggest that while children in both single-mother and single-father families initially look like they experience more behavior problems than those in two-parent families, controlling for physical and, especially, social resources explains potential differences. Similarly, when single mothers and single fathers occupy similar family environments in terms of physical and social resources, their children report similar behavior. In contrast to findings from the US, children of single mothers who occupy similar family environments as children in two-parent families in terms of resources perform slightly better in terms of externalizing behavior problems than their two-parent counterparts. We conclude that constructivist theories more accurately explain gendered parenting behavior and its consequences for child behavior problems. Environmental factors such as income, parental closeness, and participation in extracurricular activities have a significant effect on child behavior problems.

## 1. Introduction

A large and robust literature links aggressive and isolating child conduct, commonly known as child behavior problems [1,2], to less desirable child outcomes. Child behavior problems are often categorized as either internalizing problems (depression, anxiety, etc.) or externalizing problems (aggression, throwing tantrums, etc.). These behavior problems are a product of a number of factors, but primary among those factors is the child’s home environment [1]. Home environments include a myriad of factors, such as parental emotional support [2], family transitions [3,4], and family structure [1,5]. A large body of research has connected living in a single-parent family with increased behavior problems; however, conclusions from this research have often conflated number of parents available to the child with the gender of parents available to the child. In this study, we explore how behavior problems of children from two-parent households (married, cohabitating, and stepparent families) compare to children in both single-mother and single-father households in the United Kingdom and how parent gender may shape child behavior problems.

By including both single-mother and single-father families, we can explore two competing sociological theories as possible explanations for why child behavior outcomes may differ across the three groups. The first theory is the essentialist theory. Proponents of this theory suggests that there are innate, even natural, differences between men and women, and that these differences mean that fathers and mothers engage in different parenting styles [6]. Supporters of this theory would expect children in two-parent families to have fewer behavior problems compared to their peers from both single-mother and single-father families, as the latter lack access to either a female parent or a male parent. In addition, outcomes may differ across single-mother and single-father families, as the gender of the parent available in each family type would change the available family environment and, thus, child outcomes. The second theory is the constructivist theory. Supporters of this theory argue that gendered differences in parenting are a product of different gender roles and other social expectations rather than biology. While proponents of essentialist theory expect different outcomes across single-father and single-mother families because single fathers would be inherently different, proponents of constructivist theory would expect child behavior problems to be consistent between single-mother and single-father families, though perhaps worse compared to two-parent families not because of gender of available parent, but because of a lack of resources associated with having access to only one parent instead of two in the household.

To test which of these two theories is more accurate, we first establish a baseline model replicating findings from previous work comparing single-mother families to two-parent families and extend this model to include single-father families. We hypothesize in this baseline model that children from both types of single-parent families will report more behavior problems than children from two-parent families, extending previous findings about single-mother families to single-father families. We then test a model comparing children from single-mother families directly to those in single-father families, predicting that children in single-mother families will report more behavior problems when compared exclusively to single-father families because single fathers will be able to provide more resources [6,7,8,9].

### 1.1. Child Behavior Problems and Home Environment

Behavior problems in childhood are associated with a variety of negative outcomes in adulthood. Both internalizing and externalizing behavior problems in childhood are related to poor educational and occupational achievement in adulthood [10,11]. Behavior problems are also associated with higher rates of criminality among males [12] and substance use at younger ages [13]. They are even associated with an increased mortality risk [14]. As the effects of internalizing and externalizing behavior problems follow individuals through childhood and into adulthood, it is important to understand the factors that contribute to these behavior problems at a young age so that these negative outcomes can be mitigated before a child reaches adulthood.

Behavior problems seem to be especially sensitive to family environments, and vary across family structure. In the UK, children in families that have experienced more stable family trajectories experience fewer externalizing behavior problems than children from disrupted family structures [5]. Similarly, US children who live with their married biological parents experience lower levels of behavior problems than their peers from less stable family structures [7,8]. Often, the different degrees of behavior problems across family structure are closely tied to a variety of family environmental factors [1], generally divided into two categories: parental behavior and parental resources.

Evidence suggests that many of the behavioral differences between children from single-parent and two-parent households are a consequence of economic differences between the two groups [8,15]. Household income has a positive relationship to child behavioral development [16], and research on sibship size suggests that the dilution of these resources may explain why child behavior problems are more prevalent in large families [1,17]. Economic resources tend to vary across family structure, with children in two-parent households perhaps unsurprisingly having greater access to physical and human capital than children who have access to resources from only one adult in the home. Data from the United States highlight the importance of taking into account parent gender in single-parent environments, as single fathers on average have more education than single mothers, and substantially higher incomes, as well as fewer children [9]. Living with half or step siblings is negatively related to child economic well-being [18]. While we do not know as much about social capital, children in two-parent households tend to have greater access to social capital than children in single-parent households [19].

Parents play an important role in creating a positive home environment for their children. From as early as two years old, children who receive less parental emotional support are more likely to exhibit externalizing behavior problems in early childhood [2]. Among single-mother families specifically, children who feel close to their mothers are less likely to engage in antisocial behavior [20]. Parenting behavior and parent/child closeness seem to be closely related to family structure and the different roles and stressors parents may experience as part of those structures. Certain parental roles (such as having a stepchild) are associated with higher rates of depression [21], which may affect parenting behavior. Mothers who experience partnership transitions (such as dating or moving in with a new partner) experience more stress and tend to engage in harsher parenting styles [4]. These differences in parenting behavior may help explain why child outcomes vary between two-parent and single-parent households. Single parents may engage in certain parenting styles that are associated with higher levels of behavior problems. However, we know little about how these same stressors or same approaches to warm parenting operate in single-father families. Because of data limitations, researchers are often unable to study single-father families. As a result, research claiming to find relationships between parent gender and child outcomes that is derived from comparing two-parent and single-parent families may conflate the number of available parents with gender of available parent.

However, much previous research suggests that single fathers differ from their married and female counterparts in ways that may affect child behavior. For example, single fathers tend to spend less time working and more time with their children than married fathers do [22], and the relationship between parent/child closeness and child behavior problems is weaker among single-father families than it is among single-mother families [19]. Including single fathers in our analysis as a unique comparison group allows us to explore if and how parent gender affects child behavior problems.

The existing body of literature on gender and parenting is largely informed by two sociological theories: the constructivist theory and the essentialist theory. These two theories provide competing explanations of the social and biological mechanisms that produce gendered behavior. To help illustrate this, we consider the gendered differences in parenting among married couples. Mothers tend to interact more with their children on basic care and emotional training than fathers, and fathers tend to engage in more play and less childcare than mothers [23]. The essentialist theory frames these differences as innate and biological, whereas the constructivist theory frames them as a product of gendered social expectations. In order to understand which explanation is more accurate, it is important to consider not only how men and women coparent, but also how men and women behave when they are the primary caregiver. In the following sections, we further discuss how these two competing theories frame gender differences in parenting and how we expect these theories to play out in our own analysis.

### 1.2. Essentialist Perspectives on Parent Sex and Gender

The essentialist theory frames differences between fathers and mothers as a consequence of biology and early socialization [6]. Beginning at the earliest stages of both biological and social development, men and women are exposed to a variety of biological factors that encourage individuals to engage in behavior that is more masculine or feminine. These biological differences shape socialization and produce men and women who are distinct from one another in innate, gendered ways [24]. These innate differences between men and women are responsible for the different parenting approaches mothers and fathers engage in. Proponents of this theory would argue that a natural implication of this is that children must need both male- and female-type parenting in order to be properly socialized. Even if single fathers and mothers parent in similar ways (as research from the US suggests [9]), it may be that children of these families are still missing an important aspect of socialization that can only be provided by two-parent families where the two parents represent both male and female socialization. While it is perhaps surprising to find proponents of essentialist theory in western countries where women make up half of the labor market [25] and more than half of the students in higher education [26], essentialist theory continues to drive conservative policy and discussion [27].

If there are innate differences between mothers and fathers, then we assume these differences will lead to different types of child behavior in single-mother and single-father families. If the essentialist theory is accurate, we expect to see children from single-mother and single-father families exhibit different amounts of behavior problems. We also expect to see both groups score significantly worse on measures of internalizing and externalizing behavior problems compared to two-parent households, since according to this perspective, both groups are lacking essential socialization that can only be provided in two-parent households.

### 1.3. Constructivist Perspectives on Parent Sex and Gender

The constructivist theory suggests that where differences between mothers and fathers do exist, they are the consequence of different social expectations placed on men and women rather than innate gendered traits [6]. The reason why married fathers and mothers engage in different types of parenting is not because of their biology, but rather due to different social expectations around the performance of certain roles. The essentialist theory posits that parenting roles are innate and ought to be resilient to social and structural changes, but in reality, parenting roles differ across demographics and cultures and can be influenced by external factors such as workforce involvement [22,28]. This theory helps explain why single mothers and single fathers tend to engage in similar forms of parenting, at least according to US data [24].

If child behavior problems are best explained by the constructivist theory, then we expect to see very little difference in child behavior problem scores when comparing single-mother and single-father families, especially after controlling for factors such as income and parental closeness. Child behavior problems may vary between two-parent and single-parent families, but the constructivist theory frames these differences as a consequence of other social and structural factors, such as reduced access to resources from a single parent, rather than anything inherent to parent gender.

To better understand which of these theories more accurately explains the relationship between parent gender and child behavior problems, we test two hypotheses. We first test a baseline hypothesis to establish that young people in single-father families experience resource deficits that are associated with more behavior problems in ways similar to young people in single-mother families. We predict that children from both kinds of single-parent families exhibit more behavior problems than their peers in two-parent families. We then turn to a specific comparison between single-mother and single-father families to compare evidence for essentialist and constructivist perspectives of gender. We predict that children from single-mother families will exhibit more behavior problems than their peers in single-father families, but that those differences will be attenuated by controlling for the social positions of single mothers, or for income and education. It may be the case that initial comparisons suggest differences between family structures that do not persist in the face of explanations concerning mechanisms such as physical resources, social resources, family stressors, or demographic characteristics [15,16,20,29,30,31,32]. If this is the case, it would provide evidence for the constructivist perspective as men and women in similar structural positions would parent similarly. As there is evidence from previous studies that two-parent and single-parent families, as well as single-mother and single-father families, have different access to resources and different levels of exposure to stressors, and different demographic backgrounds [6,9], we take into account variables affecting each of these mechanisms. In addition, such evidence would add to research from US data and suggest a broad pattern of constructivist mechanisms. On the other hand, if we find different evidence in the UK data that we use here, that may challenge the applicability of the constructivist perspective.

## 2. Methods

### 2.1. Procedures and Participants

The data used for this study were taken from Sweep 6 of the Millennium Cohort Study (MCS), a longitudinal study that looked at more than 21,000 children born across the United Kingdom (England, Wales, Scotland, and Northern Ireland) in 2000. The Millennium Cohort Study was collected by The Centre for Longitudinal Studies at the University College London under the auspices of the National Health Service Research Ethics Committees in the South West, London, and Yorkshire, from which they received ethical approval. We obtained access to the MCS data through contract with the UK Data Services. For additional detail on the collection of the MCS, see the MCS Age 14 Survey—User Guide [33]. The original MCS sample was drawn from children born in 2000–2001; Sweep 6 was collected between 2015 and 2016 and surveyed parents and their children from across the United Kingdom. The average age of cohort members from this wave is 14. We chose Sweep 6 to maximize the number of children in our sample who had transitioned into single-father families. Our final sample included 7834 two-parent families, 2458 single-mother families, and 210 single-father families.

### 2.2. Dependent Variable

Parents were asked a series of questions about their child’s behavior from the Strengths and Difficulties Questionnaire (SDQ) [34] as part of the MCS parent interview. Following documented methods from the MCS [33], we used this questionnaire to create two scales that measure internalizing and externalizing behavior problems. The internalizing behavior problem scale is made up of 10 variables, while the externalizing behavior problem scale is made up of 12 variables. These two scales are our outcomes for this study. Our internalizing behavior scale was composed of the following survey items: target child often complains of sickness, often seems worried, often feels unhappy or tearful, is nervous or clingy in new situations, is easily scared, target child prefers to be alone, has at least one good friend, is liked by other children, is bullied by others, and gets along better with adults than children (alpha = 0.77). Our externalizing behavior scale was composed of survey items regarding the child’s outward behavior, including: does target child often have temper tantrums or have a hot temper, target child is obedient, target child usually does what adults request, does the target child often fight with other children or bully them, does the target child steal from their home, school, or elsewhere, does the target child often lie or cheat, restlessness, fidgeting, being easily distracted, thinking through decisions (reverse coded), and seeing tasks through to completion (reverse coded) (alpha = 0.76). For our regression analysis, we standardized these two scales using z-scores [5]. This allows us to compare findings for internalizing and externalizing behavior problems, which do not have the same number of indicators. Numbers above zero indicate higher than average behavior problems, and numbers below zero indicate fewer than average behavior problems. Because our dependent variables are standardized, coefficients associated with each independent variable can be interpreted as changes in standard deviations.

### 2.3. Primary Independant Variable

Ideally, we would use multifactorial measures of parent gender expression and behavior to explore how living with a single mother or a single father may affect children. Unfortunately, the MCS does not contain any such variables. As a result, in keeping with the literature comparing single-mother and single-father families in a similar way [6,9], we use parent sex as a proxy for gender. We consider three groups of families here: children who live with two parents, children who live with a single mother, and children who live with a single father. The two-parent category includes children who live with both biological parents (married and cohabiting), as well as children from families where their parent has remarried or is cohabiting with a new partner. Children in the single-mother and single-father category live exclusively with either their mother or their father; we exclude multigenerational single-parent families or other single-parent families that include additional adults. We construct the categories this way to be able to consider both number of parents and sex of available parents. This study excludes cohort members who were adopted, raised by family members other than biological parents, and twins. A very small number of families parented by two parents of the same sex were also excluded from the analyses. Children living with two parents accounted for 75% of the final sample. Children living with a single mother accounted for 23% of the sample, and children living with a single father accounted for 2% of the final sample. Two parents’ households were coded as 0, single-mother households as 1, and single-father households as 2. For each of the regression analyses we report below, we exclude the appropriate reference group. For example, in foundational analyses comparing single-father families and single-mother families to two-parent families, two-parent families are the reference category. Our final regression models exclude two-parent families entirely; in these models, single-mother families are the reference group.

### 2.4. Control Variables

Because girls often report higher levels of internalizing behavior problems, and boys often report higher levels of externalizing behavior problems [5], we control for sex of child. Sex of cohort member is recorded in the MCS; we recorded this variable as “0” for male children and “1” for female children. Previous research suggests that the closeness of the relationship between parent and child can help to reduce behavior problems in certain contexts [20]. In the MCS, children report how close they feel to both their mother and their father. The main parent respondent (usually whichever parent the child lives with, or the mother if the child lives with both parents) is asked the same question regarding how close they feel to their child. Using the responses to these questions, we were able to control for child/mother closeness, child/father closeness, and parent/child closeness. Closeness is measured on a 4-point scale, where 1 is “not very close” and 4 is “extremely close”.

Economic resources differ across family structures and often account for many of the differences between stable, two parent households and other family types [14,15,26]. We control for economic resources using the following variables: if the main parent owns a home, if the main parent receives child benefits, and the child’s household income quintile.

We control for a variety of health and substance use variables that previous research has found to influence child behavior [21,30]. This includes a score for parental depression, scores for parent and child health, and whether the main parent respondent smokes or drinks heavily. Parental depression is measured with the Kessler six-item psychological distress scale, with a range of 0–24, and is based on the frequency of depressive symptoms the main parent has experienced in the last 30 days [34]. Parent and child health are both reported by the main parent, where 0 is “excellent health” and 5 is “poor health”. Parental smoking is dichotomous: respondents receive a 1 if they smoke and a 0 if they do not. Parent drinking is also dichotomous, but we chose to mark “does drink” with a 0 and “does not drink” with a 1. It would be preferable to have more detailed measures of parental physical health behaviors, but the MCS includes only these categorical measures. We include a variable that captures child drug and alcohol use frequency where 0 represents no substance use and 4 represents frequent substance use. We control for extracurricular participation, where 0 is “no” and 1 is “yes”. We control for the number of siblings and chose to consider it as part of our parent/child health controls because we are mostly interested in the effect that this may have on child mental health [30]. Number of siblings was measured from 0 to 5 or more.

Following von Hippel [35], respondents who were missing on the variables used to create our key independent variable of family structure were dropped from our final sample. Other missing data were replaced using Stata 16’s MICE multiple imputation protocol [36] using 100 iterations. We imputed data for cohort member drug use (5.37% missing), cohort member health (3.32%), extracurriculars (3.01%), parent-reported closeness with child (3.41%), parent depression (6.91%), parent drinking (6.95%), internalizing behavior problems (3.33%), and externalizing behavior problems (3.39%). We use the 20 imputed data sets in the analyses below.

### 2.5. Analytical Strategy

Two hypotheses guide our analysis.

**Hypothesis** **1** **(H1).**
*Children from single-parent families exhibit more behavior problems than their peers in two-parent families.*


**Hypothesis** **2** **(H2).**
*Children from single-mother families exhibit more behavior problems than their peers in single-father families because of a lack of resources.*


To test these hypotheses, we used a series of 6-stepped OLS regression models. First, we ran models 1–5 using all three of our family structure categories. Then, we ran each model again, including only respondents from single-mother and single-father households. Descriptive statistics for the variables included in each model can be found in Table 1.

## 3. Results

### 3.1. Comparing Two-Parent and Single-Parent Families: Internalizing Behavior Problems

We now turn to multivariate analyses. Model 1 is a bivariate model that looks exclusively at the relationship between family structure and behavior problems. Models 2–4 include controls for other possible explanations for child outcomes. These include parent/child closeness (model 2), economic resources (model 3), and health factors (model 4). Model 5 includes the variables from each of these blocks and allows us to analyze how family structure affects child behavior problems after considering a variety of potential factors.

When comparing both single-mother and single-father families to two-parent families, Model 1 suggests that children from single-mother families score significantly higher on our internalizing behavior problems index than children from two-parent families by 0.32 of a standard deviation (*p* < 0.001). Children from single-father families do not appear to be significantly different from children with two parents. Models 2, 3, and 4 suggest a similar result—in the presence of these theoretical blocks, there is no statistically significant difference in internalizing behavior problems between young people in single-father and two-parent families, but a small and significant difference persists between single-mother and two-parent families. When parental closeness, resources, and health factors are all controlled for in Model 5, there is no longer a significant difference between either single-mother or single-father families and two-parent families. When controlling for both parental closeness and parent/child health factors, the difference between single-mother and two-parent families loses significance. Sex of child is associated with internalizing behavior; girls score an average of 0.11 standard deviations higher on the internalizing behavior problem scale than boys (*p* < 0.001). A one-point increase on the parent-reported closeness scale is associated with a 0.1 decrease (*p* < 0.001) on the child internalizing behavior scale. Similarly, poorer child health is associated with an increase in internalizing behavior problems (0.13; *p* < 0.001). Child participation in extracurricular activities also had an association with internalizing behavior; children who report participating in extracurricular activities score an average of 0.54 (*p* < 0.001) standard deviations lower on the internalizing behavior scale than children who do not. Interestingly, this model suggests that the frequency of child drug use is associated with a slight decrease in internalizing behavior problems (0.06 *p* < 0.001). Results for this analysis can be found in Table 2.

These findings provide only partial evidence for Hypothesis 1, as differences were only apparent for young people in single-mother families. When single mothers occupy similar structural positions as two-parent families—in other words, when single mothers enjoy similar levels of financial and social capital—their children exhibit similar levels of behavior problems as children in two-parent families. Surprisingly, children in single-father families exhibited no differences from two-parent families, even in the bivariate model.

### 3.2. Comparing Two-Parent and Single-Parent Families: Externalizing Behavior Problems

Table 3 presents the results of our analyses predicting externalizing behavior problems. When comparing the externalizing behavior problems of single-parent families and two-parent families, the results of Model 1 suggest that children from both single-mother and single-father families score significantly higher than their two-parent counterparts, by an average of 0.31 standard deviations (*p* < 0.001) and 0.29 standard deviations (*p* < 0.001), respectively. This pattern is consistent in Models 2 and 4 when introducing theoretical blocks for parent–child closeness and family health characteristics. However, the differences are not statistically significant when controlling for financial resources in Model 3. Instead, Model 3 suggests that differences between both kinds of single-parent families and two-parent families may be the result of income (−0.21; *p* < 0.001) and homeownership (−0.14; *p* < 0.001). The effects of income and homeownership are reiterated in Model 5, when all controls are included in the regression model. Additionally, Model 5 suggests that children in single-mother families score significantly lower on the externalizing behavior skill than children from two-parent families by an average of 0.10 standard deviations (*p* < 0.01). When controlling for all other factors, children from single-father families do not score significantly differently on the externalizing behavior scale than those from two-parent families. As was true for findings concerning internalizing behavior problems, these findings provide mixed evidence for Hypothesis 1 and are somewhat in contrast with findings from other contexts such as the United States. While there is initial support for the idea that young people in single-parent families do not cope as well as those in two-parent families, when either single mothers or single fathers occupy structural positions similar to two-parent families, especially in terms of financial resources, single fathers perform no worse, and single mothers potentially better, than two-parent families.

### 3.3. Comparing Single-Mother and Single-Father Families

The findings using the full sample call into question assumptions about the number of parents available in the home, at least in the UK context and when predicting internalizing and externalizing behavior problems, but the different results for single mothers and single fathers suggest a need to further examine the associations with parental sex. We therefore also ran the same analyses reported above comparing single-mother families exclusively to single-father families. Young people in single-mother families are the reference group. Table 4 presents results predicting internalizing behavior problems. Model 1 suggests that children from single-father families have significantly fewer internalizing behavior problems than children from single-mother families. However, these differences are small, and are rendered non-significant when we include any of our subsequent theoretical blocks. When considering parent/child closeness in Model 2, sex of child, parent-reported closeness to child, and child-reported closeness to father all have a significant association with internalizing behavior scale score, but the difference between single fathers and single mothers is no longer significant in this model. The same is true in Model 3, where both income and owning a home are associated with fewer internalizing behavior problems, but single-mother and single-father families are no longer significantly different from each other. We see a similar pattern when including family health and characteristics (Model 4); theoretical controls in this model behave as expected. Unsurprisingly, the difference between young people living with single mothers and single fathers remains non-significant in Model 5, where all controls are included. The results are similar to the results of Model 5 when using two-parent families as the comparison group. These findings are reminiscent of similar work on populations in the US [2,6,9] and provide support for Hypothesis 2. Though young people in single-father families initially appear to have an advantage, when single mothers have similar resources and structural positions to single fathers, those advantages disappear. Such results provide support for the malleability that characterizes constructivist perspectives.

The differences in externalizing behavior problems (Table 5) between children from single-father and single-mother families are not statistically significant in any of our models, including in the bivariate model (Model 1). Instead, sex of child is associated with a 0.26 standard deviation decrease (*p* < 0.001) in externalizing behavior problems, parent-reported closeness to child is associated with a 0.26 standard deviation decrease (*p* < 0.001), income is associated with a 0.11 standard deviation decrease (*p* < 0.001), and frequency of child drug use is associated with a 0.18 standard deviation decrease in externalizing behavior problem scores. While these findings do not support the initial supposition in Hypothesis 2, that children in single-mother families would start out at a disadvantage, they are reminiscent of findings for many outcomes in the literature where there are no initial differences between young people raised by single fathers and single mothers [6,9].

## 4. Discussion

### 4.1. Study Implications

Our analysis replicates many of the findings from similar studies on child behavior problems in single-parent families from both the US and the UK and adds to our knowledge about the effects of parent gender in single-parent families, particularly in the UK. We find that any behavior problem differences between children from two-parent families, single-mother families, and single-father families essentially disappear after controlling for differences in parent/child closeness, involvement in extracurricular activities, and a variety of other factors such as income and overall health, and that associations with family type are particularly sensitive to financial and social capital. Contrary to what would be expected under the essentialist theory, we do not find any differences in behavior problems between single-mother and single-father families. In fact, our results suggest that the opposite may be true. Single-mother households actually reported fewer externalizing behavior problems than two-parent households. Additionally, we find that children living with single mothers and single fathers have almost identical behavior outcomes. These findings support the constructivist theory. While certain types of family structure may be protective against externalizing behavior problems, our results provide sufficient evidence that these differences are not the result of parental gender, but rather a product of various other social and structural factors that may affect parental behavior and access to family resources.

The results also have important implications for policymakers and parents. The effect of family structure and parenting differences on child behavior is negligible compared to the effect of unequal access to financial and social resources. Our results suggest that improving access to financial resources may be the most effective way of reducing child behavior problems. Ensuring that low-income families, which may include the bulk of single-parent families regardless of parent gender, have their most pressing needs met allows parents to spend more time with their children and develop closer relationships with them. Additionally, we find that participation in extracurricular activities has consistent negative associations with both internalizing and externalizing behavior problems for children from both two-parent and single-parent families. Schools and policymakers should view the support of extracurricular activities as a low-cost, easily implemented intervention that can help reduce child behavior problems.

### 4.2. Limitations

Limitations within the MCS prevent us from analyzing children with same-sex parents, or children who do not identify with the traditional gender binary. Another limitation of our data is that they are dependent on a variety of self-reported and parent-reported measures, which may introduce some bias that we are unable to account for in the current study. Additionally, single-father families made up only 2% of our final sample, which may have reduced the statistical power. A sample with a larger number of single fathers would be at less risk of Type II error and better able to recognize potential legitimate differences between single fathers and other groups. Additionally, we have little information about whether the single fathers in the MCS data are representative of all UK single fathers. Still, given the large initial sample size, the MCS provides one of the largest samples of single fathers available for research studying youth outcomes, and one of the few samples of single fathers outside of US data. While readers should exercise appropriate caution when interpreting coefficients for smaller groups, our findings here were robust across a number of models and sensitivity tests. Examining additional family structures is beyond the scope of this paper and our focus on adjudicating between essentialist and constructivist theory; however, the approach we take here could be fruitfully applied to examining similar questions about how young people in other family structures that deviate from the white-British norm experience socioemotional development.

## 5. Conclusions

Our research illustrates the importance of studying single-father families, especially in the interest of avoiding the conflation of theories concerning number of parents and gender of parents. This is especially important given the persistence of unsupported essentialist approaches to gender in some political circles. Our findings, which support constructivist approaches, suggest that interventions focused on ensuring single parents have adequate physical and social resources will be more efficacious than those focused on parent gender [6,9,27]. Parenting and child behavior are influenced by a variety of social and structural factors and will likely differ in cultural and political contexts where parents face different social expectations and have access to different kinds of resources. Future research should explore how single-father families compare to other family structures across various cultural contexts. Our findings also underscore the importance of examining family environments outside of the often-studied United States context. Some of our findings mirror results from the US, but others show greater sensitivity to environmental factors. Taken together, our results from UK data provide additional support for constructivist perspectives on gender, broadening the case for looking at structural positions and resources to explain potential differences in child outcomes across family environments.

## Figures and Tables

**Table 1 ijerph-19-16726-t001:** Descriptive statistics.

Variable	Description	Percentage/Mean (SD)	Range
Dependent Variables			
Internalizing	Standardized score based on ten items that include the emotional and peer relationship subscales in the SDQ	0.01 (0.01)	−1.1–4.6
Externalizing	Standardized score based on fifteen items that include the hyperactivity, conduct problems, and prosocial behavior subscales in the SDQ	0.02 (0.01)	−1.2–4.6
Independent Variables			
Family Structure			
Two Parents	Child lives with both biological parents or biological parent and parent’s partner	75%	
Single Mother	Child lives with only biological mother	23%	
Single Father	Child lives with only biological father	2%	
Child Sex	Dichotomous variable, where “female” is marked as 1	50%	0–1
Father Closeness	Child reported closeness to father	2.74 (0.01)	0–4
Mother Closeness	Child reported closeness to mother	3.21 (0.01)	0–4
Parent-Reported Closeness	Main parent reported closeness to child	3.34 (0.01)	0–4
Resources			
Owns Home	Does main parent own a home?		
Yes		69%	
No		31%	
Income Quintile	Total family income grouped into five tiers		
Bottom		16%	
Second		16%	
Third		20%	
Fourth		23%	
Fifth		24%	
Receives Child Benefits	Does main parent receive child benefits?		0–1
Yes		2%	
No		98%	
Poor Parental Health	Self-reported health of main respondent	2.41 (0.01)	1–5
Parent Smoking	Does the main parent smoke?		0–1
Yes		19%	
No		81%	
Parent Drinking	Does the main parent drink?		0–1
No		77%	
Yes		23%	
Main Respondent Depression	Scale based on frequency of depressive symptoms in last 30 days	4.3 (0.04)	0–24
Sibling Number	How many siblings main child has, as reported by the main parent	1.54 (0.01)	0–7
Child Health	Main respondent reported health of child	2.52 (0.01)	1–5
Extracurriculars	Does child participate in extracurricular activities?		0–1
Yes		97%	
No		3%	
Child Drug Use	How often does child use drugs?	0.765 (0.01)	0–4

Note: Standard Deviations are in parentheses. Data are taken from Wave 6 of the Millennium Cohort Study.

**Table 2 ijerph-19-16726-t002:** Regression of family structure and internalizing behavior, parental closeness, economic resources, and parental/child health.

**Variable**	**Model 1** **Fam. Struc.**	**Model 2** **Closeness**	**Model 3** **Resources**	**Model 4** **Health**	**Model 5** **Full Model**
Single Mother	0.32 *** (0.02)	0.19 *** (0.03)	0.06 * (0.03)	0.16 *** (0.02)	−0.03 (0.03)
Single Father	0.11 (0.07)	0.17 * (0.08)	−0.11 (0.07)	−0.01 (0.07)	−0.07 (0.07)
Female		0.13 *** (0.02)			0.11 *** (0.02)
Mother Closeness		0.01 (0.01)			0.02 (0.01)
Father Closeness		−0.10 *** (0.01)			−0.07 *** (0.01)
Parent Reported Closeness		−0.13 *** (0.01)			−0.1 *** (0.01)
Owns Home			−0.15 *** (0.02)		−0.06 * (0.03)
Income Quintile			−0.13 *** (0.01)		−0.07 *** (0.01)
Receives Child Benefits			0.01 (0.07)		0.02 (0.07)
Parent Smoking				0.01 *** (0.02)	0.04 (0.03)
Parent Drinking				−0.01 *** (0.02)	−0.04 (0.02)
Poor Parent health				0.07 *** (0.01)	0.06 *** (0.01)
Parent Depression				0.06 *** (0.00)	0.06 *** (0.00)
Sibling Number				0.00 (0.01)	−0.04 *** (0.01)
Child Health				0.15 *** (0.01)	0.13 *** (0.01)
Extracurriculars				−0.57 *** (0.06)	−0.54 *** (0.06)
Child Drug Use				−0.04 *** (0.01)	−0.06 *** (0.01)

Notes: * *p* < 0.05, *** *p* < 0.001; Standard Errors in parentheses.

**Table 3 ijerph-19-16726-t003:** Regression of family structure and externalizing behavior, parental closeness, economic resources, and parental/child health.

Variable	Model 1 Fam. Struc.	Model 2 Closeness	Model 3 Resources	Model 4 Health	Model 5 Full Model
Single Mother	0.31 *** (0.02)	0.21 *** (0.03)	0.01 (0.02)	0.11 *** (0.02)	−0.10 ** (0.03)
Single Father	0.29 *** (0.07)	0.23 ** (0.07)	0.04 (0.07)	0.16 *** (0.07)	0.06 (0.07)
Female		−0.28 *** (0.02)			−0.28 *** (0.02)
Mother Closeness		−0.06 *** (0.01)			−0.03 * (0.01)
Father Closeness		−0.08 *** (0.01)			−0.04 *** (0.01)
Parent Reported Closeness		−0.27 *** (0.01)			−0.24 *** (0.01)
Owns Home			−0.21 *** (0.03)		−0.1 ** (0.03)
Income Quintile			−0.14 *** (0.01)		−0.11 *** (0.01)
Receives Child Benefits			0.01 (0.07)		0.02 (0.07)
Parent Smoking				0.19 *** (0.03)	0.12 *** (0.03)
Parent Drinking				−0.07 ** (0.02)	0.01 (0.02)
Poor Parent health				0.06 *** (0.01)	0.03 ** (0.01)
Parent Depression				0.05 *** (0.00)	0.04 *** (0.00)
Sibling Number				0.06 *** (0.01)	−0.01 (0.01)
Child Health				−0.05 *** (0.01)	0.04 ** (0.01)
Extracurriculars				−0.23 *** (0.06)	0.19 ** (0.06)
Child Drug use				0.18 *** (0.01)	0.15 *** (0.01)

Notes: * *p* < 0.05, ** *p* < 0.01, *** *p* < 0.001; Standard Errors in parentheses.

**Table 4 ijerph-19-16726-t004:** Regression of family structure and internalizing behavior, parental closeness, economic resources, and parental/child health for single mothers and single fathers only.

Variable	Model 1 Fam. Struc.	Model 2 Closeness	Model 3 Resources	Model 4 Health	Model 5 Full Model
Single Father	−0.20 *** (0.08)	−0.1 (0.1)	−0.16 (0.08)	−0.16 * (0.08)	−0.04 (0.09)
Female		0.12 ** (0.04)			0.14 ** (0.04)
Mother Closeness		−0.02 (0.03)			0.02 (0.02)
Father Closeness		−0.08 *** (0.02)			−0.06 *** (0.02)
Parent-Reported Closeness		−0.17 *** (0.03)			−0.12 *** (0.03)
Owns Home			−0.17 *** (0.05)		−0.06 (0.05)
Income Quintile			−0.13 *** (0.03)		−0.05 (0.03)
Receives Child Benefits			−0.19 (0.2)		−0.03 (0.2)
Parent Smoking				0.03 (0.05)	−0.00 (0.05)
Parent Drinking				−0.1 * (0.05)	−0.06 (0.05)
Parent Health				0.1 *** (0.02)	0.08 *** (0.02)
Parent Depression				0.05 *** (0.00)	0.05 *** (0.01)
Sibling Number				0.01 (0.02)	−0.02 (0.02)
Child Health				0.15 *** (0.02)	0.14 *** (0.02)
Extracurriculars				−0.72 *** (0.11)	−0.70 *** (0.11)
Child Drug Use				−0.03 (0.02)	−0.04 * (0.02)

Notes: * *p* < 0.05, ** *p* < 0.01, *** *p* < 0.001; Standard Errors in parentheses.

**Table 5 ijerph-19-16726-t005:** Regression of family structure and externalizing behavior, parental closeness, economic resources, and parental/child health for single mothers and single fathers only.

Variable	Model 1 Fam. Struc.	Model 2 Closeness	Model 3 Resources	Model 4 Health	Model 5 Full Model
Single Father	−0.02 (0.08)	0.00 (0.09)	0.04 (0.08)	0.05 (0.08)	0.11 (0.09)
Female		−0.27 *** (0.04)			−0.26 *** (0.04)
Mother Closeness		−0.05 * (0.02)			0.01 (0.02)
Father Closeness		−0.07 *** (0.02)			−0.03 * (0.02)
Parent-Reported Closeness		−0.33 *** (0.03)			−0.26 *** (0.03)
Owns Home			−0.2 *** (0.05)		−0.10 (0.05)
Income Quintile			−0.2 *** (0.03)		−0.11 *** (0.03)
Receives Child Benefits			−0.2 (0.2)		−0.03 (0.20)
Parent Smoking				0.13 ** (0.05)	0.06 (0.05)
Parent Drinking				−0.09 (0.05)	−0.03 (0.05)
Parent Health				0.05 ** (0.02)	0.03 (0.02)
Parent Depression				0.05 *** (0.00)	0.04 *** (0.01)
Sibling Number				0.07 *** (0.02)	0.02 (0.02)
Child Health				0.09 *** (0.02)	0.08 *** (0.02)
Extracurriculars				−0.33 ** (0.12)	−0.30 ** (0.1)
Child Drug Use				0.19 *** (0.02)	−0.18 *** (0.02)

Notes: * *p* < 0.05, ** *p* < 0.01, *** *p* < 0.001; Standard Errors in parentheses.

## Data Availability

MCS data may be acquired through contract with the UK Data Service (https://beta.ukdataservice.ac.uk/datacatalogue/series/series?id=2000031, accessed on 9 January 2022).

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
