# Peer review of "Gendered Associations between Single Parenthood and Child Behavior Problems in the United Kingdom"

_ijerph, 2022, doi:10.3390/ijerph192416726_

Round 1
Reviewer 1 Report
This is an interesting study examining internalising and externalising behaviour problems and single/two-parent families. The authors have taken advantage of a large national dataset (Millenium Cohort Study) for their sample and should definitely be congratulated on their efforts to capture what is available for single fathers.
The authors provide strong grounds for their study, with predictions from competing models. The analysis is well-planned and clear. The reporting of results and interpretation is also clear.
My only query is about inclusion of child gender as a variable. There have been many reports of gender differences in internalising and externalising including from the Millenium Cohort Study - Gutman, L.M., Codiroli McMaster, N. Gendered Pathways of Internalizing Problems from Early Childhood to Adolescence and Associated Adolescent Outcomes. J Abnorm Child Psychol 48, 703–718 (2020). https://doi.org/10.1007/s10802-020-00623-w Is it possible to consider gender in the analysis? I believe this will strengthen the article. If not, a reason for not including gender should be provided as it is an obvious gap.
There is some confusion about models and theories in the introduction. The terms seem to be used interchangeably, perhaps to give more variety in the wording, but it is important to be clear about whether a model or a theory is being used.
Line 88 – positively = positive
Line 431 – the words “When it comes to externalizing behaviour” seem redundant.
Thank you for the opportunity to review this innovative and significant manuscript.
Reviewer 2 Report
Gendered associations between single parenthood and child behaviour problems in the UK review
I think this is a worthwhile study for publication, but I think there are a number of structural and language-based issues that need to be addressed, which will also hopefully reduce replication within the manuscript, improve clarity, and make it more acceptable to readers.
Abstract
I think the term “behaviour problems” lacks specificity and sounds slightly blaming – who is it a problem for, for example? Sometimes, there is a preference to word things like this as behaviours that challenge (to reduce blame inferences) or to be more specific about the latent variable you are referring to. This comment is also pertinent for the whole manuscript.
Introduction
I struggle a little with depression/anxiety/etc being conceptualised as internalising behavioural problems, which is interesting as the primary model I practice in is predominately behavioural, however there is something about this conceptualisation that does not sit comfortably, feels a little blaming of the person with these symptoms, and takes away, possibly, from the fact that depression is characterised by lots of mood- and motivation-based difficulties, and anxiety is primarily an emotion – so the term behaviour limits the description of these. Albeit not much better, I wonder if internalising difficulties or symptoms is more accurate and less laden with subtle judgement. This, like the comment from the abstract is more of a steer, as a reader, than it is a formal direction, for the authors to consider whether their wording would be acceptable to, say, one of their participants if they were to read it.
I am a little confused why in the first paragraph of the introduction, you are describing the aims of your current study. It is more typical to report the wider context first and then build up to the more narrow focus (i.e. what your aims and hypotheses are). I’d make sure this is further down and cut from the beginning to fit with convention, as I don’t see a strong rationale for deviation here.
When explaining the essentialist model, firstly you use the word “innate” which makes me as the reader thing you are referring to this be nature-based differences in parenting. It may be the model does say this, as I am not entirely familiar with it, but if the model doesn’t, I’d change your wording so the reader does not misinterpret. Secondly, you state that a typically-heterosexual family made up of both a male and female parent, which balancing both of the “innate” differences in parenting style, will mean a child is less likely to develop difficulties, however I feel unclear how? Like, why would having a balancing of these two prototypical parenting styles mean better outcomes? I think a sentence or two needs to be added to explain this hypothesis.
Now reading the next section, I think the essentialist model does focus on nature over nurture, so it may be my comment above about this is redundant.
When introducing the constructionist model, I am again unclear of how the hypothesised outcomes for the child would be different – even if the parenting differences were due to nature or social construction/nurture, why would this mean that the outcome is different for the child? I think this needs explaining more as it feels rather unclear and good to presume your reader has little background knowledge in the area you are writing about.
Just to note again that before getting to section 1.1, you are again reintroducing what you do in your study which is outside convention for an empirical paper. This should be reserved for the end of your introduction unless there is good reason to state it earlier. This is because the introduction often follows a structure where the reader is introduced to the broad “issue” you are covering and then you slowly narrow the focus on what your specific “issue” is, until eventually selling to the reader where the gaps are and how your aims fill a gap somewhere.
I’m confused about the hypothesis that states single mother families will result in more “problem children” because of lower resources – this risks sounding prejudice at best and misogynistic at worst – please back anything like this up with research to show this is indeed the case for the cohort you are focusing on, i.e. is there evidence to show single mother families have lower resources than single father families?
Confusingly, around line 61, you are starting to state your results and conclusions – I think this is grossly misplaced in the structure. The introduction does not typically have any reference to your findings, as this is reserved for your results and discussion section. I would move these to the appropriate section.
Please define what a “more stable family structure…”, cited in line 79, is? Same in line 82. The language does somewhat convey negative judgement towards families that do not have two married parents or similar, so would again just prompt the thought for you as a writer “would my participants be happy with the language used in this paper”? I don’t think I would like my family being described as less stable if it was merely that I was a single parent.
I see in the paragraph starting at line 86 that you begin explaining some of the things I asked for above – this is fine, but I think the explanation needs to happen at the first mention of the idea/relationships/variable. This paragraph generally looks good to me though, as it reads as a lot less judgmental and more linked to the evidence-based
From line 115, you start to re-address your study aims etc, I think this needs to go at the end of the introduction as mentioned above.
Line 118 and the following few lines make statements that I think need to be referenced, as “we know” is not sufficient in academic papers without citation and anecdotal knowledge is not usually accepting without good justification to say why you cannot cite something.
Line 128 onwards is starting to repeat information in a way and I wonder if it could be integrated with the above sections.
In line 149 you state that differences in parent styles by gender mean you need both parents to be socialised effectively as a child – this again feels very presumptuous that both styles are necessary for health development – I would try and explain why the theory thinks this is true and also maybe name that this could be controversial. Just on a general note – I am surprised this model is accepted currently, as I find it to rather judgemental of family structures that do not fit the typical structure from the 1950s and that it doesn’t look like tones of evidence supports it (although I may be wrong in that assumption). I can however now see why this model may suggest poorer outcomes for children if there are not two parents of difference genders as it assumes that both styles are necessarily for “healthy development”, reading between the lines.
From line 173, it also now makes sense why you hypothesise that under a constructivist model, that there would not be differences in outcomes based on parent structure – this is why it is important to not state your hypotheses or aims, and especially your findings, until the end of the introduction so the reader can use all the information about the context first to inform how they interpret the rationale behind your hypotheses.
Generally, based on the above comments, I think your introduction needs restructuring, which hopefully will also reduce your repetition (but do check that too).
Method Section
Your procedures and participants section is actually just your participant section – the procedure section needs to detail how data was collected to a replicable standard.
Please cite the SDQ.
I am unclear why you needed to make a Z score for the SDQ scores within your regression analysis, please explain? I don’t feel clear why you’d need to establish what was above or below average in this case rather than using raw scores?
Usually, the variables are detailed in the “Design” section alongside what the research design is, rather than in separate sections as they are presented here. As with above, if there is a good rationale to deviate from the convention, then this is okay but I can’t see why that would be the case here.
Why are you controlling for variables (line 254) that have no evidence of influence outcomes? It seems an interesting decision to include more variables than needed from a family-wise error and power point of view, even if you do have a large sample. Please justify this decision.
Please detail the outcome measures used for the parent variables, like depression, as I assume they didn’t just use their own items in data collection but a validated measure? If they did use their own items without psychometric development, then this is a significant limitation as you cannot be sure the questions measure the latent variables.
As you used MICE procedures, do this mean that the outputs of the analyses are all pooled using Rubin’s rules or some other variation of how you present the collated outputs? Also, for MICE, how many iterations were used to make your 20 datasets? Finally, you say you deleted or “dropped” incomplete data listwise for some, but imputed via MICE for others. It would seem odd to use listwise deletion or even pairwise deletion if using MICE? It would be good to explain this decision making.
I am unclear why your hypotheses are in your analytic strategy section – these typically go in your introduction at the end as mentioned above.
There is no mention that I could see about your ethical approval. You say the data collection was covered by an ethics board, but was your study specifically?
Results
There is lots of information in the results section that belongs in the discussion section. The results is simply to clear state what you found, what was statistically significant and what was not, and what the effect sizes were. Whereas, I see bits of discussing how the results fit with ideas introduced in the introduction and your hypotheses, which should be in the discussion.
Discussion
Discussions rarely, if ever, start with limitations. These come at the end of the discussion just before conclusions and just after implications. Please revise the structure to fit convention.
Line 410+ - it is a bit of an over stretch to say you may have found differences if the sample was bigger – instead it is more accurate to say that power may have been reduced in particular groups, thus it may have precluded potential differences increasing a type 1 error.
Why does a smaller sample introduce bias? I can infer why, but you need to explain why you think so.
There are many other limitations not cited: exclusion of family structures that deviate from the white-british norm, reliance on self-report measures or parent-report measures of childrens’ difficulties, etc. Please expand.
This section needs to pull down the discussion in the results section currently and generally expand on how the findings relate to the wider literature, etc. I think tidying up and clarifying things in your introduction will also help doing so in your discussion so that it is clear why you findings refute or support the models.
Reviewer 3 Report
Thank you for the opportunity to review “Gendered associations between single parenthood and child behavior problems in the United Kingdom”. This manuscript has a number of strengths, including the use of a large sample, the incorporation of a focus on single fathers in addition to single mothers/2-parent households, the theory-based approach, and the effort to account for a number of potential environmental confounds.
Despite these strengths, I do have some concerns about the manuscript in its current form that I have described below.
Introduction
- The authors do a nice job of introducing the two competing theories that are the central focus of their analysis. However, I believe the introduction would benefit from a clearer introduction of the various covariates that are ultimately considered in their model. For example, the results are broken up into a series of regression models based on four environmental categories that are progressively integrated into the model (i.e., 1) parent-child relationship quality/closeness, 2) family resources, 3) parent and child health, and 4) family stress). I think it would more clearly guide the reader if the introduction included a separate section for each of these areas to support understanding for why these factors would be considered to be important for internalizing/externalizing symptoms, and how they relate to essentialist/constructivist theory.
- The primary research questions and a brief summary of the results are provided early in the introduction (pg. 2 lines 52–67). I recommend instead presenting these research questions/hypotheses at the end of the introduction and in the analytic plan. Additionally, it may be more appropriate to provide a summary of the results in the discussion rather than the introduction.
- Gender and sex seem to be used interchangeably throughout the paper. However, these are distinct constructs. Please be consistent in the terminology used throughout and ensure that it is consistent with how this variable is operationalized/measured in the sample.
Methods/Measures
- Why were 5 items used for the internalizing subscale and 9 items for the externalizing subscale as the SDQ typically recommends 10 items each for the internalizing and externalizing subscales.
- Please report the internal consistency of all composite variables/scales included in the current study.
- The authors state “two parents’ households were coded as 0, single-mother households as 1, and single-father households as 2” (pg 5, lines 236–237). In the regression models, please clarify if this variable was recoded using dummy coding to ensure that the interpretation of the categories was appropriate to the regression framework (i.e., if two-parent household was treated as the reference var, there would be 2 dummy coded variables in each model with the single-mother household var coded 0: not a single-mother household, 1: single-mother household, and with the single-father household var coded 0: not a single-father household, 1: single-father household. Currently, the procedure surrounding this is unclear.
- Given the way that this variable is coded, I recommend the authors consider renaming the “parent health” variable to something like “poor parental health”. Or recode this variable so that 5 is “excellent health” and 0 is “poor health”.
- The family stress variable seems to share some significant overlap with other covariates in the model based on the items used to operationalize this construct (e.g., parent substance use, economic resources, parent mental health). Do intercorrelations between the parent substance use and economic resources items from the health indicators and family stress indicators suggest problematic overlap, or that these reflect different constructs? If it is the former, I recommend the authors consider consolidating some of these variables further, or removing the items that share overlap so that they are not over-represented in the model.
- It would be helpful to better understand the degree of missing data present (e.g., % of missing data for each variable, if missingness is associated with any of the study variables, etc.) in order to better inform the generalizability of the results.
Results
- Please report the intercorrelations among the study variables included in the regressions. It may work well to convert Table 1 into an intercorrelations table by removing the description column (and ensuring these details are reported in the manuscript text) and including a row or column for the descriptives of percentage/mean and SD. Further, in addition to percentage/mean and SD, please report the min, max, and n for each variable where appropriate.
- Did any multicollinearity concerns emerge in any of the models?
- Reporting the effect sizes for the detected effects would help improve interpretability of the results.
- Pg. 8, line 327–328, should this sentence instead read as: “single-father families exhibited no differences from two-parent families, even…”
Discussion
- I recommend moving the limitations to the end of the discussion and instead leading with the important findings/implications of the results.
- Why was child sex/gender not considered in the current study? Given that parent sex/gender is a central focus of the paper, one thing that I continually came back to as I was reading was what role child sex/gender might be expected to play, if any. What do essentialist and constructivist theories suggest in this area? Perhaps the potential role that child gender/sex plays can be expanded upon in the introduction and/or discussion.
Minor comments:
- Please very carefully proofread the manuscript for typos. I caught several:
o pg. 2 line 88, should be “positive” instead of “positively”
o pg. 2 line 93, should be “than” instead of “that”
o pg. 4. Line 190 should insert “it” in front of “will”
o pg. 11 line 344 should be “statistically significant” instead of “significantly significant”
o pg. 14 there are typos in line 363
Round 2
Reviewer 3 Report
I found the authors to be highly responsive to the reviewer comments. I feel that the manuscript is significantly improved in its current form and I have no further comments. I commend the authors on a strong revision and an interesting manuscript.